# A Hybrid Dragonfly Algorithm for Efficiency Optimization of Induction Motors

**DOI:** 10.3390/s22072594

**Published:** 2022-03-28

**Authors:** Niraj Kumar Shukla, Rajeev Srivastava, Seyedali Mirjalili

**Affiliations:** 1Department of Electrical Engineering, Shambhunath Institute of Engineering and Technology, Prayagraj 211015, India; 2Department of Electronics and Communication, J K Institute of Applied Physics and Technology, University of Allahabad, Prayagraj 211002, India; rajeev_jk@rediffmail.com; 3Centre for Artificial Intelligence Research and Optimization, Torrens University Australia, Brisbane, QLD 4006, Australia; ali.mirjalili@gmail.com; 4Yonsei Frontier Laboratory, Yonsei University, Seoul 03722, Korea

**Keywords:** induction motor, PI controller, speed control, dragonfly algorithm, group search optimizer, particle swarm, optimization, algorithm, meta-heuristic

## Abstract

Induction motors tend to have better efficiency on rated conditions, but at partial load conditions, when these motors operate on rated flux, they exhibit lower efficiency. In such conditions, when these motors operate for a long duration, a lot of electricity gets consumed by the motors, due to which the computational cost as well as the total running cost of industrial plant increases. Squirrel-cage induction motors are widely used in industries due to their low cost, robustness, easy maintenance, and good power/mass relation all through their life cycle. A significant amount of electrical energy is consumed due to the large count of operational units worldwide; hence, even an enhancement in minute efficiency can direct considerable contributions within revenue saving, global electricity consumption, and other environmental facts. In order to improve the efficiency of induction motors, this research paper presents a novel contribution to maximizing the efficiency of induction motors. As such, a model of induction motor drive is taken, in which the proportional integral (PI) controller is tuned. The optimal tuning of gains of a PI controller such as proportional gain and integral gain is conducted. The tuning procedure in the controller is performed in such a condition that the efficiency of the induction motor should be maximum. Moreover, the optimization concept relies on the development of a new hybrid algorithm, the so-called Scrounger Strikes Levy-based dragonfly algorithm (SL-DA), that hybridizes the concept of dragonfly algorithm (DA) and group search optimization (GSO). The proposed algorithm is compared with particle swarm optimization (PSO) for verification. The analysis of efficiency, speed, torque, energy savings, and output power is validated, which confirms the superior performance of the suggested method over the comparative algorithms employed.

## 1. Introduction

Nowadays, asynchronous motors are massively used in the process of industrial applications with the requirement of a larger performance. From the IM [1,2] discovery period, it is assumed as the actuator dominance with a constant speed tactic. Within several decades, this has been deployed in numerous industries as drivers or actuators for generating mechanical forces and motions. Every year in industries, the operational units of IM are increasing; hence, the enhancement of efficiency is a top priority for industries by minimizing the losses which will increase as a result of global conservation of energy. Some of the challenging characteristics of IM, specifically the cage IM over DC motor, are less cost, small inertia, less maintenance, high efficiency, simpler design, inherently self-starting, and absence of collector brush systems. However, there are some demerits in IM, which is complex in nature, including a multivariable mathematical model, nonlinearity, and intrinsically incompetent offering of flexible speed operation. The aforesaid issues can be solved by constructing the motor control operators with smart scalar and vector strategies.

The control approach has assumed that the reference magnetic flux [3,4] is stable and nearer to its rated level (λ_rnm_) on behalf of larger dynamic performance in spite of operating points. The resultant outcome of such procedures has unacceptable energy efficiency, while the IM is assumed as under-loaded. Most of the research explorations have demonstrated that about 45% of IMs drive 40% of their rated load. To tackle this challenge, the auto-adjustment of flux has to be made online regarding the load torque for attaining the reduced power loss. Actually, the flux leakage value with unique optimal on every operating point is tracked by the efficiency optimization algorithm derived [5,6,7] and pertained it to controlled IM [8,9]. For optimizing the drive system’s efficiency, many techniques have been evolved in the literature. They are categorized generally into two major models: MBO and SAO.

The MBO has possessed some challenges, which is its larger sensitivity to parameter variations and the count of arithmetic operations contained within the loss model’s solution. The characterization of SAO in [10,11] is made by its parameter insensitivity. However, it poses some promising challenges such as very slow convergence and high torque ripples to the optimal operating point while distinguished over MBO. In such cases, it has a very large search space and thereby outcomes in large time in the quest for optimal operating conditions. In fact, the optimization algorithm in [12,13] may underestimate the optimal flux with no adaptation of real-time parameters’ mechanisms, which may as well direct the destabilization of the drive system and impact the problem of “motor stalling”, particularly in the existence of an immediate change in load torque. It should be noted that in the full operating region, there is a critical issue in the identification of IM parameters accurately [14,15,16] and simultaneously, even by using the IM parameter stacking algorithms. Over the last few years, researchers have been working on evolutionary algorithms [17] such as genetic algorithms (GA) and swarm-intelligence-based algorithms [18] such as PSO, GSO, DA, FF, and ACO for optimization purposes under different working environments. These meta-heuristic algorithms are very efficient and determine an accurate estimation of optimal solutions compared to conventional algorithms.

This framework has introduced a novel model to maximize the efficiency of induction motors. For this, the objective model is constructed using the speed-control induction motor drive with the tuning of the PI controller. In this PI controller, the optimal tuning of PI controllers such as proportional gain and integral gain is made by introducing a newly hybrid algorithm named SL-DA, which is the hybridization of DA and GSO. The rest of the paper is organized as follows: Section 2 explains the review on the literature of induction motor and the methodology that is discussed. Section 3 presents the proposed model of AC sensor-less speed control of the induction motor drive. Section 4 presents the original DA, GSO, and PSO along with the proposed SL-DA algorithm. The experimental results are discussed in Section 5. Finally, Section 6 concludes the research work and suggests future directions.

## 2. Literature Review

The challenges and the features of traditional induction motor models are described in this section. Various algorithms are deployed for the efficient working of IM. Some of them are explained in the following. An SVM-based DTC strategy for IM [19] has presented for better optimization of IM energy and has minimized flux ripples and torque and further has constant switching frequency and good waveform of the stator current. A better dynamic and larger robustness has been ensured by this nonlinear approach against external disturbance. However, there are disadvantages of variable switching frequency and increased control difficulty at low-speed regions. A DFOC method [20,21] was proposed that maximizes electric energy saving and has better control of rotor flux and motor speed. However, it has poor sensitivity and parameter variation, and controller complexity is increased. 

Adaptive hybrid LMT by Farhani et al. [22] poses enhanced motor efficiency and has a maximum energy saving, but the power-saving potential is mainly based on robustness and accuracy of flux regulation. A flux search controller [23] has improved the efficiency and accuracy of loss calculation. On the basis of adaptive gradient descent of motor flux value having quick response and easier implementation, the flux search controller was implemented. The implemented model was presented with the combination of a suitable loss method of a six-phase induction motor. However, it needs additional hardware and has slow convergence and torque variations. In 2017, Kong et al. [24] proposed a harmonic current injection method that defines better efficiency and torque density and has increased power density; still, it is complex in nature. The analysis on the impact of harmonic currents on yoke flux density and air-gap flux density has been made altogether. The implemented model has wholly made the utilization of optimal coefficients for equivalent harmonic current, and multiple control freedoms were extracted. The torque density has enhanced the nine-phase induction motor by 60% by sustaining the similar stator current density with the implemented model. The construction of three-phase induction motor was completed for validating the efficiency of multiphase motors. In 2015, Laamari et al. [25] presented a narrative model on the basis of grouping of the extensive Kalman filter with PSO for evaluating the rotor and speed flux of an induction motor drive and suggested that PSO has precise estimation and faster optimization. Further poses better performance in load torque and rotor resistance. However, the scattering problems cannot be exploited and are critical to explain the initial design parameters. The effectiveness of the implemented model has been evaluated using the simulation experiment on rotor and speed flux estimation. 

In 2018, Zeb et al. [26] exploited several narrative, robust, and adaptive control mechanisms, called (a) FLC on the basis of LMA, (b) FLC on the basis of SDA, (c) FLC on the basis of NA, and (d) FLC on the basis of GNA for the IVC 3 phase IM. FLC has less power consumption and voltage dips. Further, it has a fast dynamic response and is robust to parameter uncertainties. Despite this, it lacks real-time response, has low speed, and takes a longer time to run. In 2018, Costa et al. [27] suggested that ACO has better performance, and the retuning of controllers is not required. Nevertheless, it lacks secondary information usage and has slow sequential processing. This approach has been taken within the DTC-SVM control loops that include stator flux linkage, computation of the linkage stator flux, electromagnetic torque, and rotor speed. In Table 1, we summarize the main contributions in the field of induction motors for optimizing efficiency.

### Proposed Methodology

Induction motors are considered the industries’ workhorse because of their efficiency, power/mass relation, low cost and quite maintenance-free operation in their life span. Nonetheless, the motors that work on low efficiency waste more energy, which in turns maximizes the operational cost. As for the consequence of a large count of operating units and large energy consumption, even a small rise in improvement of efficiency has had a considerable effect on operational cost and the whole energy consumption. To make this possible, this proposal aims to introduce a newly hybrid algorithm that hybridizes the concept of dragonfly algorithm (DA) and group search optimization (GSO). DA is a newly meta-heuristic optimization technique [28] that solves the single-objective, discrete, and multi-objective problems. GSO is the algorithm [29] that is inspired based on animal searching behavior, which solves the continuous optimization problems. Furthermore, for a large amount of numerical testing, Mirjalili suggested that DA executes in a better way as compared to GA and PSO [30]. Conventionally, in [31,32], the authors used the key attributes of loss model control (LMC) as well as search control (SC) together for assessment and reproduction of optimal flux component of current (I_ds_), for optimal efficiency operation of IM. However, the model struggles with the performance rate when there is variation in the load profile. Hence, to make a solution for this, this proposal introduces a new contribution that keeps the (I_ds_) value constant and optimizes the current regulator by optimally tuning the proportional integral (PI) controller as depicted in Figure 1, by improving the PI controller parameters such as proportional gain and the integral gain as well. This optimal selection obviously paves way for the optimal efficiency operation of induction motors.

## 3. Model of AC Sensor-Less Speed Control of Induction Motor Drive

### General Block Representation

The system consists of a pure DC supply, which needs to be converted into AC by employing an Insulated Gate Bipolar Transistor (IGBT) inverter. An IGBT-based 3-phase inverter is used in this paper, which consists of twelve IGBT switches, out of which four IGBT switches per phase are arranged in an H-Bridge configuration. The scheme is depicted in Figure 2. A three-phase asynchronous machine is implemented within the asynchronous machine block, which is termed as single squirrel-cage, wound rotor, or double squirrel-cage. Here, the operation is made in motor mode where motor torque and speed are varied under different conditions. For the conventional method, IM uses the vector control technique where the voltage, current, and torque of a 3-phase induction motor are controlled by time-varying components. The speed is used as a feedback loop in a PI controller at different values of load torque. The proposed speed controller (AC) block exemplifies a PI speed regulator technique for AC machines that is utilized within vector-controlled drives. Here, two operating instances are present: the initial one is having both the outputs of torque as well as the flux references and the second has only the output of the torque reference. The speed of the machine is given in rpm as N; the speed reference of the machine is in rpm and is denoted as $N^{*}$. In this, the machine’s torque reference is termed as input, when the parameter of regulation type is fixed as torque. The flux reference is symbolized as $Flux^{*}$ and the unit of this is weber. The induction motor speed is controlled by tuning the PI controller; i.e., the proportional and integral gain of the PI controller is fine-tuned via a proposed SL-DA scheme, which is deeply explained in Section 4. 

## 4. Proposed Efficiency Optimization in Induction Motor

### 4.1. Optimized Speed Control

The speed control concept is depicted in Figure 3. Here, the quadrature current is expressed as $I_{q}$, the direct current is given as $I_{d}$, the reference quadrature current is exemplified as $I_{q}{}^{*}$, the reference direct current is specified as $I_{d}{}^{*}$, and the reference three-phase current is delineated as $I_{abc}{}^{*}$. The PI controller [33,34] is considered as a renowned controller which poses the ability to maintain the exact set points. The operational modeling of PI controller is on the basis of grouping of two controller modes viz. proportional and the integral mode. The logical expression for PI controller is gained from $k_{p}$ and $k_{i}$ parameters. The common form of PI controller is stated as per Equation (1), where $k_{p}$ is the proportional gain and $k_{i}$ is termed as integral gain.
(1)$$u(t)=k_{p}e(t)+k_{i}\int e(t)dt$$

Using the Laplace transform, Equation (1) is changed to Equation (2).
(2)$$U(s)=k_{p}E(s)+k_{i}\frac{E(s)}{s}$$

PI regarding the time constraints is expressed in Equation (3).
(3)$$U(s)=k_{p}\left[1+\frac{1}{\tau_{i}}\right]E(s)$$
where $k_{i}=\frac{k_{p}}{\tau_{i}}$ and $k_{p}=k_{d}/\tau_{d}$.

The proportional and integral terms provide fine control for the error signal and minimize the steady-state errors of the model. For a closed-loop control system, when precise tuning of *k_p_* and *k_i_* values are performed, then there is a small decrease in the rise time and improvement in settling time [34]. Many software-based concepts have been realized for the tuning purpose of the controllers. The main contribution or novelty of the work is involved in the speed-control block, where the optimized PI controller is newly implemented to enhance the controlling performance. In Figure 4, the two gains $k_{p}$ and $k_{i}$ of the PI controller inside the flux PI block are optimized using the proposed optimization algorithm named SL-DA.

### 4.2. Objective Function with Gain Encoding

During the optimization of gains in the PI controller inside the speed-control block, the fixed objective function is to maximize the efficiency of the induction motor, which is as per Equation (4). The computation formula for the efficiency of induction motor is based on Equation (5).
(4)obj=max(η)
(5)$$\eta =\left[\begin{array}{l} Speed/S_{p}\\ Tem/T_{m} \end{array}\right]$$

In Equation (5), *Speed* refers to the output rotor speed, *S_p_* indicates the reference speed, Tem denotes the electromagnetic torque, and $T_{m}$ refers to the load torque. As mentioned earlier, the above objective function is attained by optimizing or tuning the gains $k_{p}$ and $k_{i}$ of the PI controller.

### 4.3. Procedure for Gain Update

For maximizing the efficiency of the induction motor, a procedure is fixed here, which depends on the fine tuning of PI controller gains. Figure 5 represents the flowchart of the proposed methodology. Initially, the load torque $T_{m}$ and reference speed $S_{p}$ are set. Further, the initialization of gains $k_{p}$ and $k_{i}$ is performed, and the rotor speed Speed and electromagnetic torque Tem are calculated. With these collected parameters, the induction motor’s efficiency is calculated. The efficiency is considered as the objective function, and the solution is updated with $k_{p}$ and $k_{i}$ using the proposed SL-DA algorithm. The parameters are continuously updated along with the solution update until the termination is reached, which in turn provides the best solution.

### 4.4. Conventional DA Algorithm

Across the world, 3000 diverse species of dragonfly are found, and their lifecycle contains two main stages: nymph and adult. Most of their lifecycle is comprised of the nymph stage; after that, the dragonfly becomes adult by metamorphism. The static as well as dynamic swarming behaviors of dragonflies have been mimicked in DA [28]. Dragonflies work as small clusters in the static swarm (exploration) stage and fly to and from inside a closer area for prey hunting. The core aspects of a static swarm are the abrupt changes and local movements within the flying path. The dynamic swarms (exploitation) have an enormous count of dragonflies for making the swarm migrate over long distances in a particular direction. Moreover, survival is the core objective of any swarm and hence the entirety of the swarm’s individuals can be drawn to food sources and divert to the external enemies. Based on the above-mentioned two behaviors, the position update of swarm individuals is of five types, namely: alignment, control cohesion, attraction (over food sources), distraction (over external enemies), and separation. Equation (6) denotes the separation of the ${\hat{i}}^{th}$ dragonfly and is denoted as $D_{\hat{i}}$ to form its neighbor.
(6)$$D_{\hat{i}}=-\sum_{\hat{j}=1}^{NI^{\prime }}\left(Z^{\prime }-{Z^{\prime }}_{\hat{j}}\right)$$

In this, the current individual’s position is given as $Z^{\prime }$, the ${\hat{j}}^{th}$ neighboring individual’s position is denoted as ${Z^{\prime }}_{\hat{j}}$, and the count of neighboring individual is stated as $NI^{\prime }$. Equation (7) estimates the alignment of such dragonflies. Here, the velocity of ${\hat{j}}^{th}$ neighboring individual is delineated as ${v^{\prime }}_{\hat{j}}$. The calculation of cohesion is depicted as per Equation (8). Equation (9) explains the estimation of attraction over the food source. In this, the food source position is specified by Fd. Equation (10) shows the computation of distraction over the external enemies. In this, the position of the enemy is given by Eny.
(7)$$Al_{\hat{i}}=\frac{\sum_{\hat{j}=1}^{NI^{\prime }}{v^{\prime }}_{\hat{j}}}{NI^{\prime }}$$
(8)$$Ch_{\hat{i}}=\frac{\sum_{\hat{j}=1}^{NI^{\prime }}{Z^{\prime }}_{\hat{j}}}{NI^{\prime }}-Z^{\prime }$$
(9)$$Fs_{\hat{i}}=Fd-Z^{\prime }$$
(10)$$EE_{\hat{i}}=Eny+Z^{\prime }$$

The dragonflies’ behavior is considered on the basis of these five corrective patterns. In search space, the position of dragonflies’ update as well as the movements of their simulation are made using two vectors, namely position $Z^{\prime }$ and step $\Delta Z^{\prime }$. The step vector is the same as the PSO’s velocity vector, and on the basis of this PSO algorithm, the development of the DA algorithm is made. Equation (11) has demonstrated the step vector which controls the movement of dragonflies.
(11)$$\Delta {Z^{\prime }}_{t+1}=(d^{\prime }D_{\hat{i}}+a^{\prime }Al_{\hat{i}}+c^{\prime }Ch_{\hat{i}}+f^{\prime }Fs_{\hat{i}}+e^{\prime }EE_{\hat{i}})+\delta .\Delta {Z^{\prime }}_{t}$$
where the separation weight is denoted as $d^{\prime }$, the ${\hat{i}}^{th}$ individual separation is given by $D_{\hat{i}}$, the alignment weight is depicted as $a^{\prime }$, the alignment of ${\hat{i}}^{th}$ individual is delineated as $Al_{\hat{i}}$, the cohesion weight is $c^{\prime }$, the ${\hat{i}}^{th}$ individual cohesion is $Ch_{\hat{i}}$, the food factor is given as $f^{\prime }$, the food source of ${\hat{i}}^{th}$ individual is expressed as $Fs_{\hat{i}}$, the enemy factor is indicated by $e^{\prime }$, the ${\hat{i}}^{th}$ individual enemy position is portrayed as $EE_{\hat{i}}$, the inertia weight is symbolized with δ, and the iteration count is defined as t. The position vector estimation is as per Equation (12).
(12)$${Z^{\prime }}_{t+1}={Z^{\prime }}_{t}+\Delta {Z^{\prime }}_{t+1}$$
where t expresses the current iteration. The dragonflies need to roam over a search space in a random walk manner (Levy flight) for improving the stochastic behavior, randomness, and exploration, when having no neighbor solution. At this stage, the updated position of the dragonflies is made using Equation (13).
(13)$${Z^{\prime }}_{t+1}={Z^{\prime }}_{t}+Levy(h)\times {Z^{\prime }}_{t}$$
(14)Levy(h)=0.01×ran1×Φ|ran2|1l
(15)Φ=(Γ(1+ξ×sin(πξ2)Γ(1+ξ2)×ξ×2(ξ−12))1ξ
(16)Γ(z)=(z−1)!

The dimension of position vectors is expressed by h, the two arbitrary numbers within the range 0 and 1 are denoted as $ran_{1}$ and $ran_{2}$, and ξ is assigned to be a constant value. The steps as well as the position of each and every dragonfly within every iteration are upgraded by using Equations (11)–(13). The dragonfly identification is made through assessing the Euclidean distance between the total dragonflies and selects $NI^{\prime }$ for updating $Z^{\prime }$ and $\Delta Z^{\prime }$ vector. At this position, the upgradation process is repeated up until the final criteria is met. The pseudo-code representation of the conventional DA model is given in Algorithm 1.
**Algorithm 1:** Pseudo code of conventional DA algorithmInitiate the dragonfly population using $Z_{i}(i=1,2,\ldots n)$
Start off the step vectors $\Delta Z_{i}(i=1,2,\ldots n)$
While the termination criteria is not achieved
Compute the objective values of all dragonflies Upgrade the enemy as well as food source Upgrade d,a,c,f, and e
 Estimate D,Al,Ch,Fs, and EE using Equations (6)–(10) Upgrade the neighboring region radius When the dragonfly constitutes a least-neighbor dragonfly  
The velocity vector is upgraded per Equation (11) 
The position vector is upgraded per Equation (12) Else
 
The position vector is upgraded per Equation (13) End if
 The new locations are validated and corrected according to different boundariesEnd while

### 4.5. Conventional GSO Algorithm

The GSO algorithm with a population cluster is called a group, and every individual within the population called a member [24]. Over a search space of n− dimensional, the $p^{th}$ member at $l^{th}$ searching iteration (bout) provides a present position $Z_{p}^{l}\in {ℜ}^{n}$, a head angle $\phi_{p}^{l}=(\phi_{p_{1}}^{l},.....,\phi_{p_{\left(n-1\right)}}^{l})\in {ℜ}^{n-1}$. The $p^{th}$ member’s search direction is the unit vector $F_{p}^{l}(\phi_{p}^{l})=(f_{p_{1}}^{l},\ldots \ldots ,f_{p_{n}}^{l})\in {ℜ}^{n}$, which is computed from $\phi_{p}^{l}$ through a polar to a Cartesian coordinate transformation, which is stated in Equations (17)–(19).
(17)$$f_{p_{1}}^{l}=\prod_{k=1}^{n-1}\cos(\phi_{p_{k}}^{l})$$
(18)$$f_{p_{q}}^{l}=\sin(\phi_{p_{\left(q-1\right)}}^{l}).\prod_{k=q}^{n-1}\cos(\phi_{p_{k}}^{l})\;\;\;(q=2,\ldots ,n-1)$$
(19)$$f_{p_{n}}^{l}=\sin(\phi_{p_{\left(q-1\right)}}^{l})$$

In the GSO algorithm, there are three members’ kinds in a group: scroungers and producers having behaviors on the basis of PS model, and the dispersed one which evaluates the motion of random walk. While taking the optimization issues, the open patches are the ones that are assigned to be unknown optima and that are arbitrarily distributed within a search space. Therefore, the searching of patches is made by moving within the search space by the group members. In accordance with the relevant phenotypic features, the producer and the scroungers are considered to be the same; hence, the two roles can switch with each other. 

In iteration, a group of members that is located in the more challenging area and is presented as the premier fittest value is selected as producer. After that, it halts and scrutinizes the whole environment for seeking the optima (resources). Scanning is considered as a core module within the search direction. In GSO algorithm, the behavior of the producer $Z_{g}$ at the $l^{th}$ iteration is given as follows:

Step (1): The scanning will be made by producer at 0° and after that scan laterally by arbitrarily sampling three points within the scanning field: one point within 0°, given as per Equation (20); one point at the right side of hypercube, given in Equation (21); and one point in the left hand of the hypercube, stated in Equation (22).
(20)$$Z_{x}=Z_{g}^{l}+rn_{1}v_{\max}F_{g}^{l}(\phi^{l})$$
(21)$$Z_{rn}=Z_{g}^{l}+rn_{1}v_{\max}F_{g}^{l}(\phi^{l}+rn_{2}\theta_{\max}/2)$$
(22)$$Z_{v}=Z_{g}^{l}+rn_{1}v_{\max}F_{g}^{l}(\phi^{l}-rn_{2}\theta_{\max}/2)$$

Here, the arbitrarily distributed random number having mean 0 and std. deviation 1 is denoted as $rn_{1}$, and the sequences are distributed uniformly within the interval (0,1) which is stated as $rn_{2}$.

Step (2): The best point will be determined by the producer with the best resources, i.e., fitness value. When the best point is provided with better resources rather than the current position, then the member will move over this point. Otherwise, the members will remain in their present position and twist their heads toward a novel generated angle at random. In Equation (23), the maximum turning angle is denoted as $\alpha_{\max}\in {ℜ}^{1}$.
(23)$$\phi^{l+1}=\phi^{l}+rn_{2}\alpha_{\max}$$

Step (3): The heads will be turned to zero degree if the producer is not able to determine a better area after the a teration. In Equation (24), $a\in {ℜ}^{1}$ is a constant.
(24)$$\phi^{l+a}=\phi^{l}$$

At the time of every spell on searching, the group member count is selected as scroungers. The opportunities for joining the resources found by the researchers will be searched by these scroungers. This GSO algorithm only takes the area-copying feature, which is assigned as the basic scrounging behavior. The modeling of the area-copying behavior of $p^{th}$ scrounger on the $l^{th}$ iteration has depicted as a random walk over the producer and is stated in Equation (25).
(25)$$Z_{p}^{l+1}=Z_{p}^{l}+rn_{3}\circ (Z_{g}^{l}-Z_{p}^{l})$$

Here, the uniform random sequence in the interval (0,1) is depicted by $rn_{2}\in {ℜ}^{n}$. The Hadamard product or the Schur product is given as ‘∘’, where the entry-wise product of two vectors is calculated. 

The ranging is happened, when the $p^{th}$ group member gets dispersed, and this is called disperse member rangers. Naturally, the searching concepts are performed by the ranging animals that involve systematic search strategies and random walks for locating the resources effectively. The rangers only deploy the random walks and are given in Equation (26).
(26)$$v_{p}=a.rn_{1}v_{\max}$$
and move to a new point that is stated as per Equation (27).
(27)$$Z_{p}^{l+1}=Z_{p}^{l}+v_{p}F_{p}^{l}(\phi^{l+1})$$

The conventional GSO algorithm pseudo code is explained in Algorithm 2.
**Algorithm 2:** Conventional GSO algorithmAssume l=0
Initiate position randomly Zp and head angles ϕp of entire group membersCompute the fittest values of initial members f(Zp)
while (the stopping criteria not met) for (every member p in the group)  Select producerDetermine the producer Zp of the group members  Executeproducing(1) The producer will examine at 0° and then scrutinize laterally by arbitrarily sampling three points in the scanning field using (20) to (22)   (2) Discover the finest point with the foremost resource (fittest value). If the finest point shows a more preferable resource than its present location, then it will move at this point. If not, it will remain in its present location and move its head toward a new angle using Equation (23)   (3) When the producer cannot determine a preferable area after an iteration, it will move its head back to 0° using (24)  Execute scroungingTo carry out scrounging, select 80% from the remaining members   Execute dispersionFor the remaining members, they will be distributed from their present location to execute ranging: (1) produce an arbitrary head angle using (5) and (2) select a random distance vp from the Gauss distribution using (8) and proceed towards a new point using (9)  Calculate fitnessCompute the fittest value of the present member f(Zp)
 End for assume l=l+1
End while

### 4.6. Conventional PSO Algorithm

In 1995, Kennedy and Eberhart [35] introduced a bio-inspired algorithm known as particle swarm optimization (PSO), which was based on the swarm intelligence technique. In this algorithm, a flock of birds moves in a group to navigate and forage. Each particle is a vector of values for the decision variables of the problem that gets updated in each iteration using a velocity vector. The velocity vector considers the current velocity of the particle, movement toward personal best (*pbest*), and tendency toward the global best (*gbest*). The flowchart of the PSO algorithm is depicted in Figure 6.

### 4.7. Proposed SL-DA Algorithm

DA has achieved much attention among researchers from diverse fields because of its simplicity [36,37]. However, it needs improvement in internal memory as it directs to premature convergence to local optima [38]. Similarly, GSO is the well-known population-based optimization technique that is based on the inspiration of animal group living theory and searching behavior. The main shortcomings in this algorithm are its poor exploration and exploitation ability and slower convergence. 

In order to alleviate the above-mentioned drawbacks, this paper proposes a new optimization algorithm called the Scrounger Strikes Levy-based dragonfly algorithm (SL-DA), which incorporates the concept of GSO within the DA algorithm [39]. The proposed framework is detailed as follows: in the conventional DA algorithm, if the condition dragonfly poses, at least one neighbor dragonfly is satisfied; the velocity vector and position vector update are exploited as per Equations (11) and (12), respectively. Otherwise, the update of the position vector is handled using the Levy update in Equation (13). In this proposed model, the same condition is checked and if satisfied, the velocity vector and position vector are made the same as the conventional DA model. In the else condition, the scrounger behavior of the GSO in Equation (25) is carried out instead of the Levy update in the conventional model. The proposed SL-DA algorithm’s pseudo-code is explained in Algorithm 3. The proposed SL-DA algorithm’s flowchart is illustrated in Figure 7.
**Algorithm 3:** Pseudo code of proposed SL-DA algorithmInitiate the dragonfly population $Z_{i}(i=1,2,\ldots n)$
Start off the step vectors $\Delta Z_{i}(i=1,2,\ldots n)$
While the termination criteria are not met Compute the objective values of whole dragonflies Upgrade the enemy as well as food source Upgrade d,a,c,f, and e
 Estimate D,Al,Ch,Fs, and EE using Equations (6)–(10) Upgrade the region of neighboring radius When the dragonfly constitutes a least-neighbor dragonfly   Update velocity vector as per Equation (11)  Update position vector as per Equation (12) Else   Update position vector as per the scrounger behavior of the GSO in Equation (25) End if   The new locations are validated and corrected according to different boundariesEnd while

The suggested SL-DA algorithm has the advantage of precise estimation and good accuracy of loss calculation. It supports global and local search capabilities. The convergence rate is better with maximum exploitation capabilities. On the other hand, if we consider sequences of random decisions, then it is not dependent.

## 5. Results and Discussion

In this section, the results of the proposed method are presented, discussed, and analyzed. The simulation work of the proposed model was evaluated using MATLAB 2018a. The performances of the proposed method were examined in terms of power efficiency, speed, and torque under four configurations such as speed 90 and torque 130, speed 120 and torque 150, speed 150 and torque 180, and speed 180 and torque 200. Further, the analysis of the adopted model was exploited over the other conventional models in [31] such as vector operation and optimal operation regarding efficiency, energy-saving, and input power. The analysis between the proposed SL-DA method and PSO algorithm are also presented and an analysis of efficiency, output power, and savings in energy has been presented in different tables. 

### 5.1. Analysis on Power Efficiency

Figure 8 explains the analysis of the power efficiency of the proposed model over conventional models. This analysis is made over four configurations by varying the time. In fact, the power efficiency is on the basis of speed and torque and is assigned to be in maximum for the effective operation of the system. Initially, the efficiency of power is started with a low value and achieves the maximum over the time seconds. On comparing the conventional models in [31] with this proposed work, the results are discussed in the following paragraphs. 

Inspecting Figure 8a, the proposed work with speed 90 and torque 130 at time 1.6 has achieved maximum power efficiency, which is 25.77% better than vector operation. Similarly, in Figure 8b, at time 2.2 with speed 120 and torque 150, the implemented model has attained better performance than both optimal operation and vector operation by 9.52% and 4.02%, respectively. Subsequently, the proposed work is compared over other models for other speeds and torques by varying the time, and the resultant outcome has demonstrated the betterment of the proposed work with increased power efficiency. 

Another analysis on efficiency has been performed between the proposed SL-DA model with the PSO model to show that the proposed method has high accuracy. The results are depicted in Figure 9, where we can see that exploration of SLDA is high as compared to PSO.

### 5.2. Analysis on Speed

Figure 10 delineates the analysis of the speed of the proposed model against the conventional model under four configurations by varying the time seconds. In this analysis, the speed is set to some value initially. For instance, in Figure 10a, the speed is fixed as 90. The system performance seems to be evaluated efficiently only when the fixed speed is attained on simulation. For instance, on considering the same Figure 10a, the simulation speed attains the fixed speed of 90, by which the system performance is better achieved. Likewise, for the other configurations, the simulation speed of the proposed model is attained with better value, i.e., fixed value, and thus proves the improved performance of the system with better efficiency.

The resulting analysis of the speed graph of the proposed SL-DA algorithm is compared with the PSO and depicted in Figure 11. It can be clearly seen that the SL-DA algorithm reduces the overshoot and attains stability more conveniently as compared to the PSO algorithm. 

### 5.3. Analysis on Torque

Figure 12 exhibits the performance analysis of torque of the proposed model against conventional models by varying times under four configurations. In this, the torque is fixed with some value, and if that value is attained on simulation, then the overall system performance is considered to be effective. Initially, the torque value is at maximum and is decreased linearly over time. The results thus obtained in this way are explained as follows.

As per Figure 12a, the implemented work with speed 90 and torque 130 at time 2.2 has achieved the targeted value and is better than other conventional models’ optimal operation and vector operation by 43.82% and 43.97%, respectively. Similarly, from Figure 12d, the proposed model with speed and torque as 150 and 180 at time 3.65 has achieved the same fixed value 150 and that is 18.8% and 20.69%, respectively. The simulation of the implemented model thus validates the improved system performance with better attainment of torque.

Analysis of torque graph under four configurations has been performed between the proposed SL-DA and PSO models. The results can be seen in Figure 13. From the obtained results, we can observe that there is more smoothness in the torque graph, and ripples are slightly minimized. Hence, the proposed SL-DA algorithm shows better results as compared to the PSO algorithm.

### 5.4. Analysis on Efficiency

Table 2 describes the efficiency analysis of the proposed work over conventional and PSO models under four configurations. The efficiency of the proposed SL-DA working with speed 90 and torque 130 is achieved with better performance and shows a 13.47%, 8.13%, and 5.29% improvement for optimal operation, vector operation, and PSO model, respectively. Similarly, the implemented method in terms of efficiency with speed 150 and torque 180 has achieved superiority over the optimal operation and vector operation by 0.63%, 13.27%, and 1.34%, respectively. From the overall efficiency analysis, it is demonstrated that the suggested SL-DA model has attained supremacy over the other conventional model as well as PSO model with better efficiency.

### 5.5. Analysis on Energy Saving

Table 3 explains the energy-saving analysis of the proposed SL-DA model over conventional models and PSO models under four configurations. The energy-saving table is assessed based on the efficiency in Table 2, and the assessment is conducted as per Equation (28). On comparing the overall performance of the model on energy saving, the proposed model seems to attain better performance than the conventional and PSO models. There is an improvement in energy saving of induction motor drive when operated with SL-DA algorithm.
(28)$$\%\;energy\;\;saving=\frac{proposed-conventional}{conventional}*100$$

### 5.6. Analysis on Output Power

Table 4 delineates the analysis of the output power of the proposed work over conventional models and PSO under four configurations. The output power obtained by the proposed SL-DA work is increased as compared to the conventional models’ optimal operation, vector operation, and PSO algorithm in speed 90 and torque 130 by 13.47%, 8.13%, and 5.29%, respectively. Subsequently, the proposed model with speed 180 and torque 200 has achieved improved performance in output power as compared to the optimal operation, vector operation, and PSO by 13.44%, 8.23%, and 6.07%, respectively. From the simulation, it is observed that the suggested algorithm has more effective performance with increased output power than other conventional models. 

### 5.7. Obtained Proportional and Integral Controller Gain Values by Conventional Method and Proposed SL-DA Algorithm 

Table 5 defines the values of $k_{p}$, $k_{i}$, and efficiency for different configurations of speed and torque that were obtained optimally using the conventional vector control, optimal operation, and PSO algorithm along with the proposed SL-DA algorithm by varying the torque and speed. The optimal values obtained by the SL-DA algorithm tune the proportional−integral controller more optimally as compared to other approaches, which gives better performance.

The simulation results of the conventional vector model, optimal operation, PSO algorithm, and proposed SL-DA optimization method are presented in this paper, and a comparative analysis among all the methods has been performed. On comparing the simulated results and the data presented in the tables, it is observed that the SL-DA optimization concept has shown superiority over other conventional methods as well as on the PSO algorithm. The proposed SL-DA concept has shown improvement in the performance of induction motor drive in terms of efficiency and energy-saving with an increase in output power. The proposed SL-DA method provides a fine-tuning of the PI controller, which improves the results of induction motor drive.

## 6. Conclusions

This research work proposed a new contribution for maximizing induction motor efficiency. The objective model was obtained from the speed-control induction motor drive, where the PI controller was tuned. Further, the PI controller’s gains such as proportional gain and integral gain were optimally tuned by deploying a new hybridized algorithm named SL-DA, which incorporates the concept of DA with GSO. The proposed SL-DA results are tested and compared with the PSO algorithm for verifications. It is observed that the exploration capability of SL-DA is higher than PSO. As a result, the SL-DA algorithm outperformed PSO and other conventional methods. Finally, the performance of the suggested model was distinguished over other models in terms of controlling the analysis of torque and speed, the efficiency and energy-saving analysis of drive, and analysis on improvement in output power, and thus validates the superiority of the proposed work. For future work, other meta-heuristic algorithms can be implemented to tune the proportional–integral controller for minimizing loss, enhancing efficiency, and evaluating the performance of induction motor drive.

## Figures and Tables

**Figure 1 sensors-22-02594-f001:**
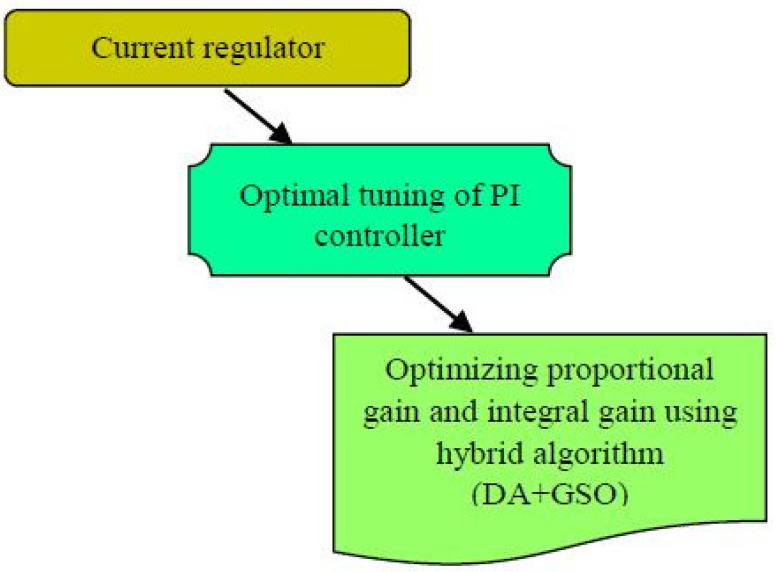
Proposed efficiency optimization model of induction motors.

**Figure 2 sensors-22-02594-f002:**
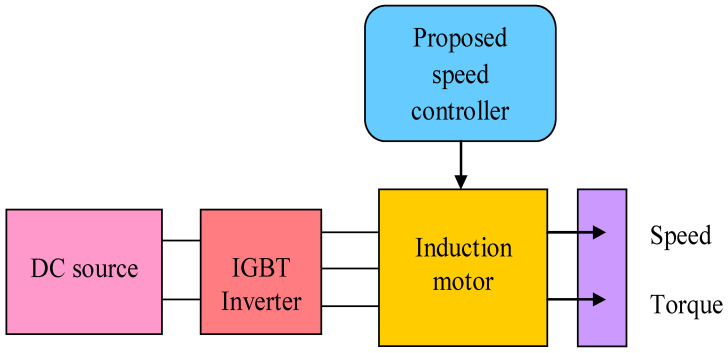
Representation of proposed model.

**Figure 3 sensors-22-02594-f003:**
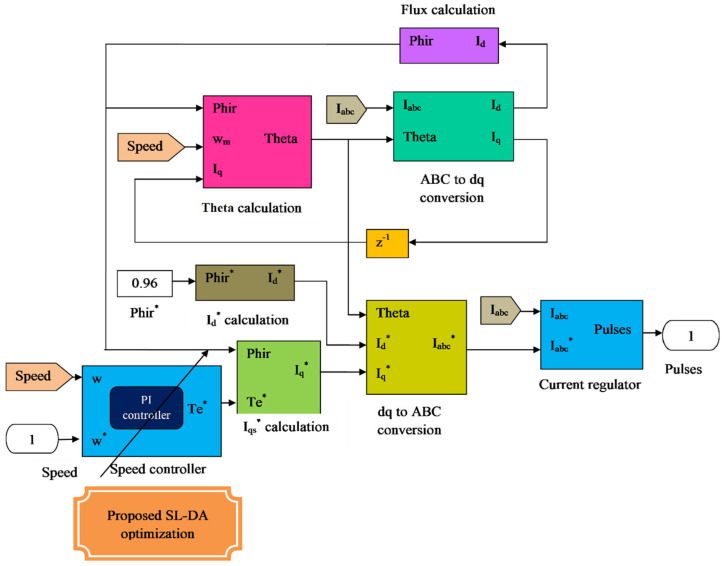
Systematic representation of SL−DA based speed-control block.

**Figure 4 sensors-22-02594-f004:**
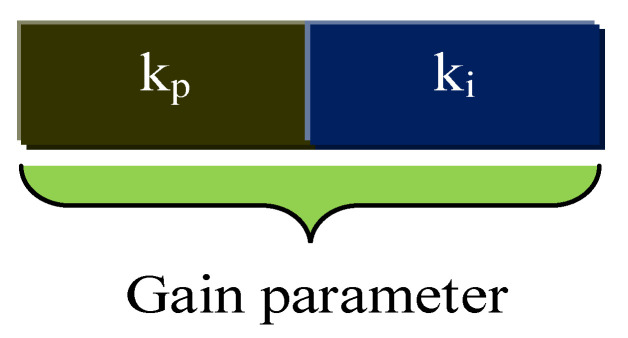
Solution encoding of proposed algorithm.

**Figure 5 sensors-22-02594-f005:**
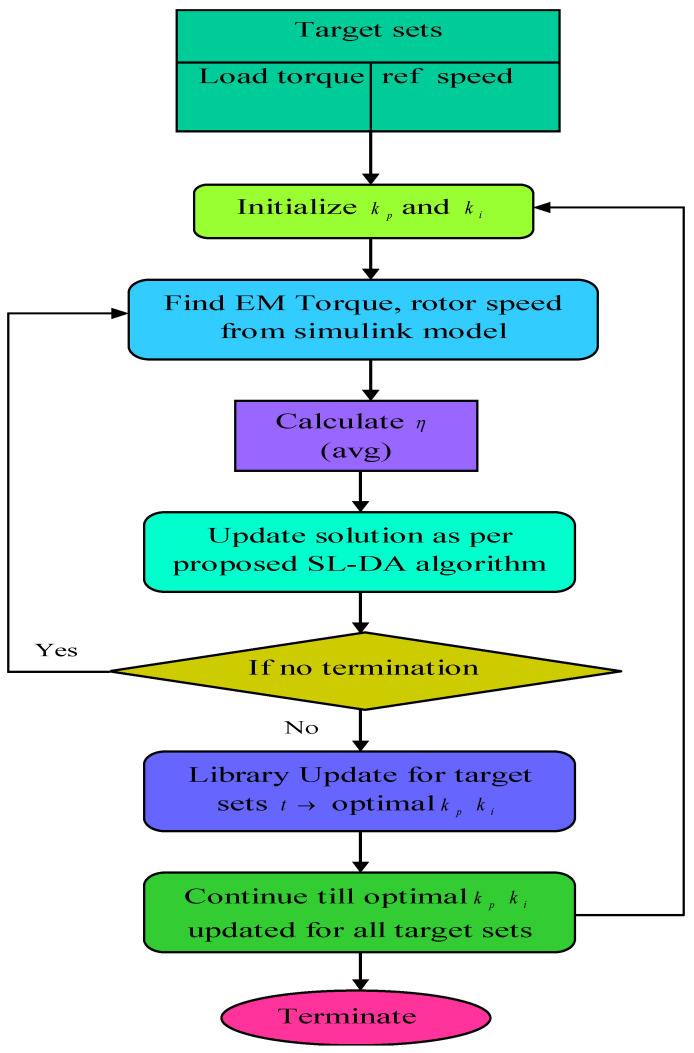
Procedure adopted for gain update in PI controller of speed-control block.

**Figure 6 sensors-22-02594-f006:**
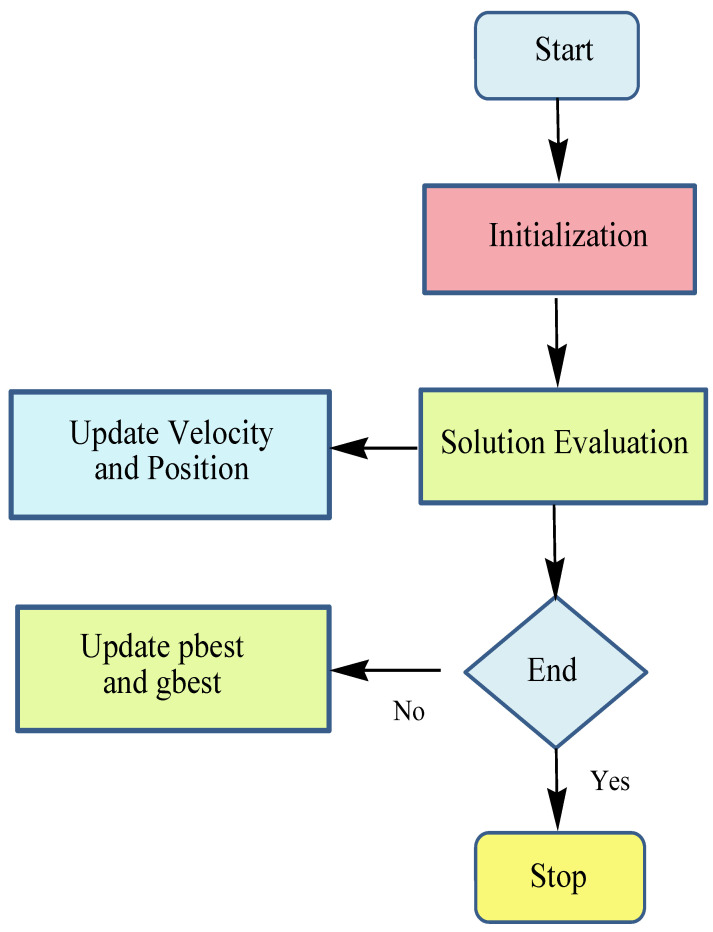
Flowchart of PSO algorithm.

**Figure 7 sensors-22-02594-f007:**
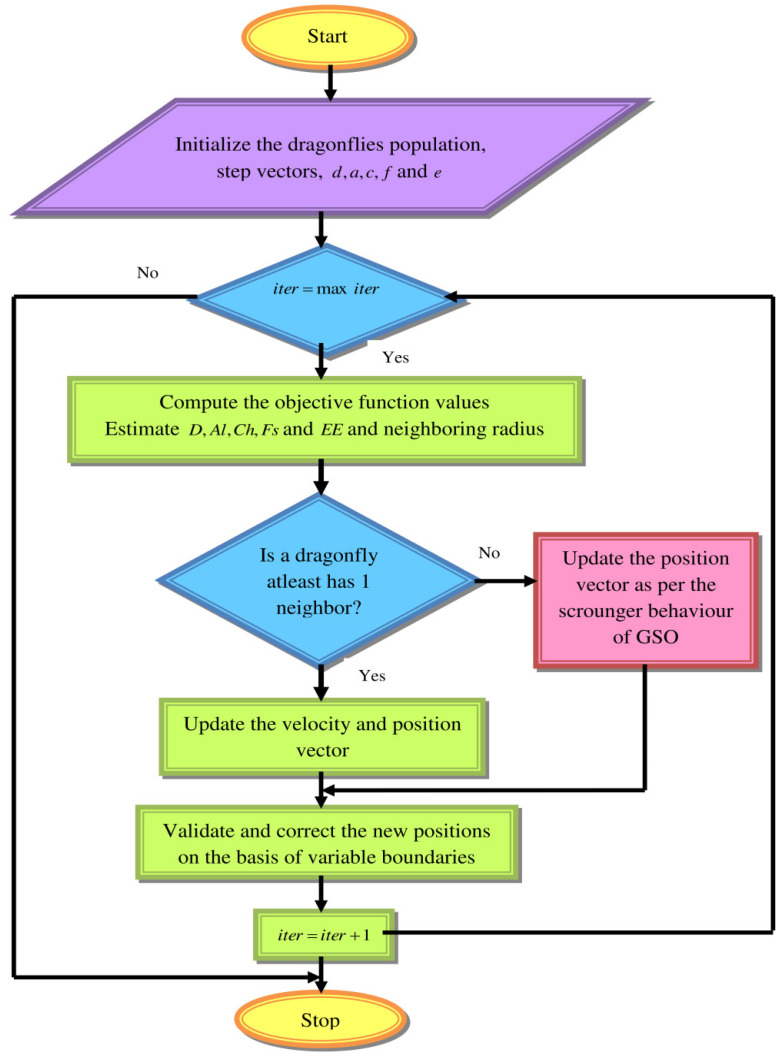
Flowchart of proposed SL-DA algorithm.

**Figure 8 sensors-22-02594-f008:**
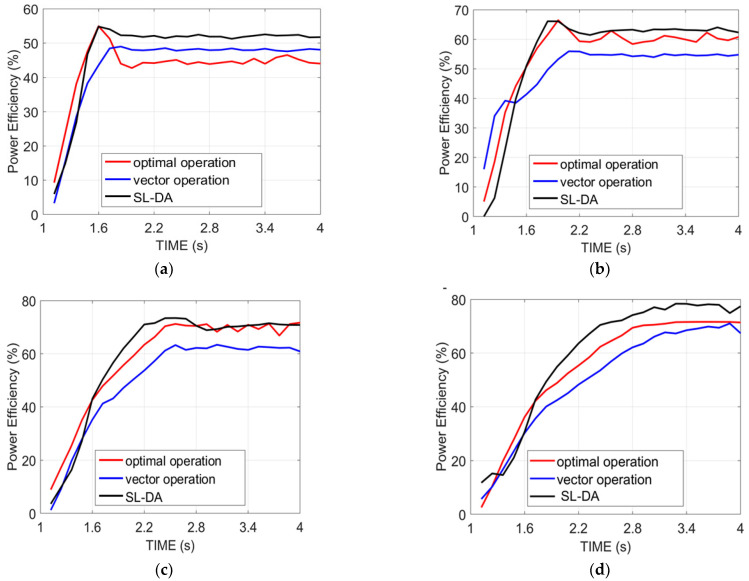
Analysis on the power efficiency of proposed work over conventional models: (**a**) speed 90 and torque 130; (**b**) speed 120 and torque 150; (**c**) speed 150 and torque 180; and (**d**) speed 180 and torque 200.

**Figure 9 sensors-22-02594-f009:**
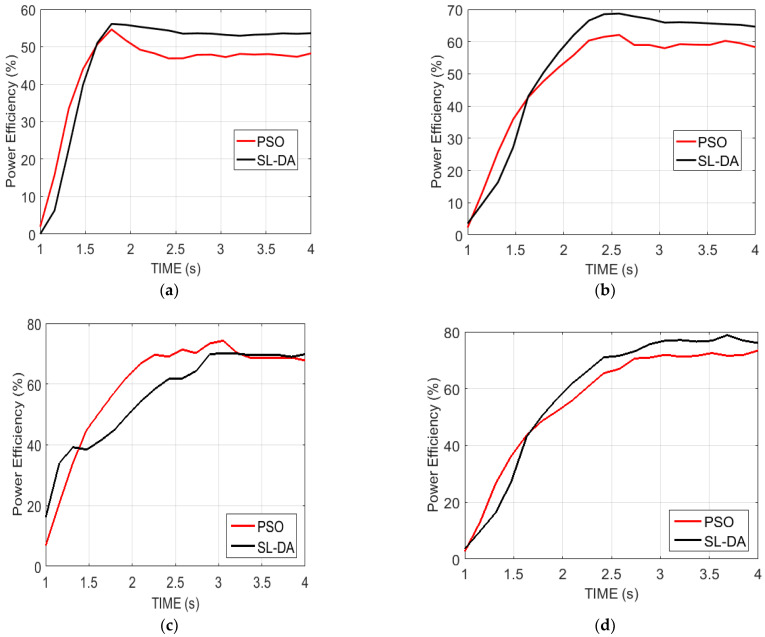
Analysis on the power efficiency of the proposed work over PSO: (**a**) speed 90 and torque 130; (**b**) speed 120 and torque 150; (**c**) speed 150 and torque 180; and (**d**) speed 180 and torque 200.

**Figure 10 sensors-22-02594-f010:**
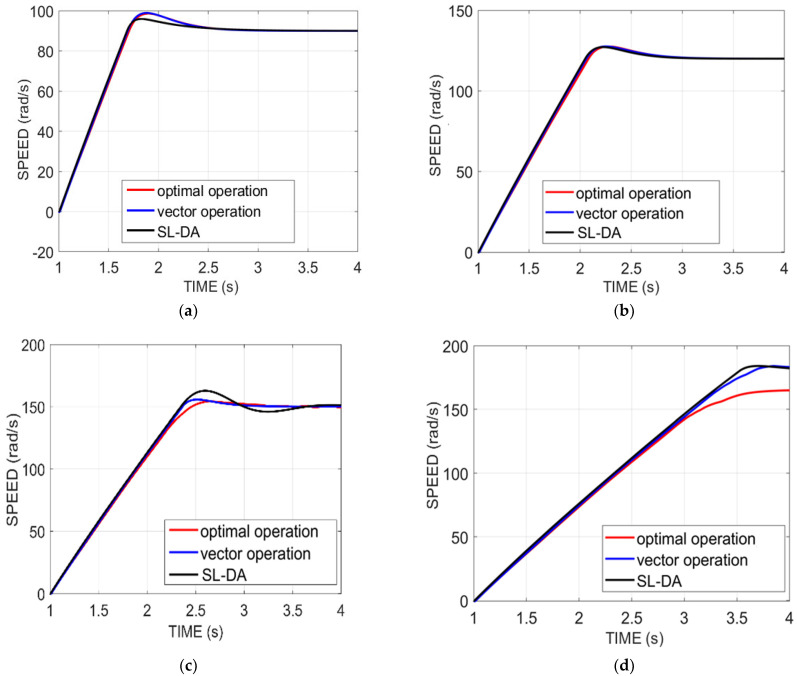
Speed analysis of proposed work over conventional models: (**a**) speed 90 and torque 130; (**b**) speed 120 and torque 150; (**c**) speed 150 and torque 180; and (**d**) speed 180 and torque 200.

**Figure 11 sensors-22-02594-f011:**
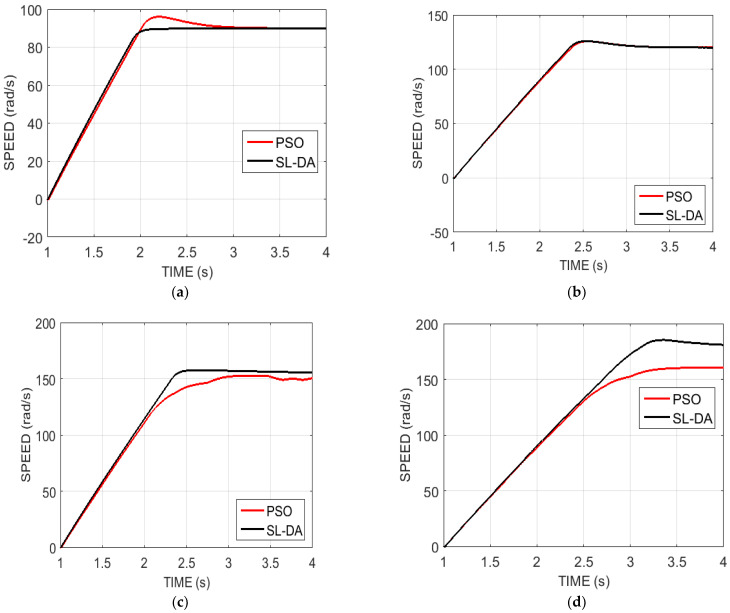
Speed analysis of proposed work over conventional models: (**a**) speed 90 and torque 130; (**b**) speed 120 and torque 150; (**c**) speed 150 and torque 180; and (**d**) speed 180 and torque 200.

**Figure 12 sensors-22-02594-f012:**
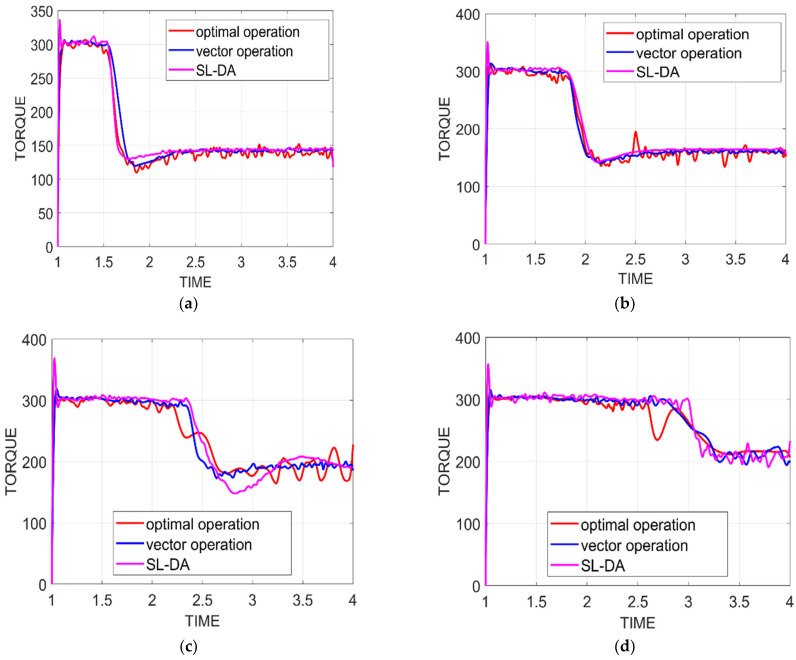
Torque analysis of proposed work over conventional models: (**a**) speed 90 and torque 130; (**b**) speed 120 and torque 150; (**c**) speed 150 and torque 180; and (**d**) speed 180 and torque 200.

**Figure 13 sensors-22-02594-f013:**
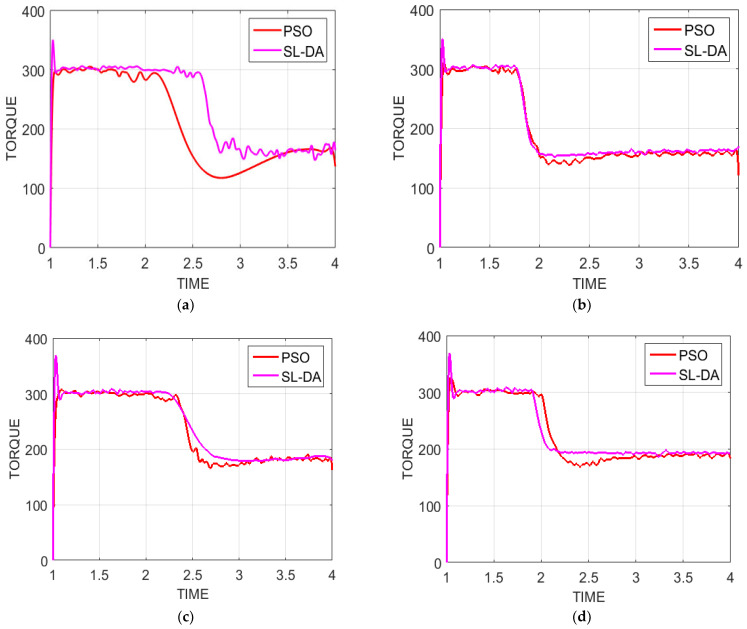
Torque analysis of proposed work over PSO: (**a**) speed 90 and torque 130; (**b**) speed 120 and torque 150; (**c**) speed 150 and torque 180; and (**d**) speed 180 and torque 200.

**Table 1 sensors-22-02594-t001:** Methodology used in traditional induction motor models.

Methodology	References
SVM-DTC	Ammar et al., (2017) [20]
DFOC	Farhani et al., (2017) [21]
Adaptive Hybrid LMT	Farhani et al., (2017) [22]
Flux Search Controller	Taheri et al., (2012) [23]
Harmonic Current Injection Method	Kong et al., (2017) [24]
Particle Swarm Optimization	Laamari et al., (2015) [25]
Fuzzy Logic Controller	Zeb et al., (2018) [26]
Ant Colony Optimization	Costa et al., (2018) [27]
Dragonfly Algorithm	Jafari and Chaleshtari (2017) [28]
Group Search Optimizer	He et al., (2009) [29]
Hybrid Dragonfly Algorithm	Wang et al., (2021) [30]
Neural Network	Choudhary et al., (2015) [31]
ANFIS Controller	Shukla et al., (2020) [32]
Proportional Integral Controller	Idi et al., (2015) [33]

**Table 2 sensors-22-02594-t002:** Efficiency analysis of proposed work over state-of-the-art models.

	Efficiency (%)			
	Vector Operation	Optimal Operation	PSO	SL-DA
speed = 90 and torque = 130	45.818	48.08	49.38	51.99
speed = 120 and torque = 150	54.602	60.253	59.208	63.121
speed = 150 and torque = 180	62.174	69.986	69.493	70.426
speed = 180 and torque = 200	67.997	71.273	72.719	77.139

**Table 3 sensors-22-02594-t003:** Energy saving analysis of proposed work over state-of-the-art models.

	Comparing Optimal Operation with Proposed SL-DA Work	Comparing Vector Operation with Proposed SL-DA Work	Comparing PSO with Proposed SL-DA Work
speed = 90 and torque = 130	8.1323%	13.471%	5.285%
speed = 120 and torque = 150	4.759%	15.602%	6.608%
speed = 150 and torque = 180	0.6287%	13.273%	1.342%
speed = 180 and torque = 200	8.2303%	13.445%	6.078%

**Table 4 sensors-22-02594-t004:** Output power analysis of proposed work over state-of-the-art models.

	Input Power (kw)	Output Power (kw)			
		Vector Operation	Optimal Operation	PSO	SL-DA
speed = 90 and torque = 130	22.273	10.205	10.709	10.998	11.58
speed = 120 and torque = 150	28.292	17.047	15.448	17.713	17.858
speed = 150 and torque = 180	35.149	24.6	21.854	24.426	24.754
speed = 180 and torque = 200	40.118	28.593	27.279	28.956	30.946

**Table 5 sensors-22-02594-t005:** Obtained proportional and integral values of conventional method and proposed SL-DA algorithm.

	Vector Control			Optimal Operation			PSO Algorithm			SL-DA Algorithm		
	*k_p_*	*k_i_*	Efficiency	*k_p_*	*k_i_*	Efficiency	*k_p_*	*k_i_*	Efficiency	*k_p_*	*k_i_*	Efficiency
speed = 90 and torque = 130	20.24	32.21	45.818	20.61	33.20	48.08	19.8	34.41	49.38	21.68	34.76	51.990
speed = 120 and torque = 150	12.10	28.09	54.602	13.12	28.71	60.253	19.22	29.38	59.208	13.70	30.60	63.121
speed = 150 and torque = 180	13	26	62.174	13.71	27.33	69.986	14.09	25.94	69.493	14.33	26.21	70.426
speed = 180 and torque = 200	2.06	12.03	67.997	2.93	14.16	71.273	3.15	14.77	72.179	3.37	15.96	77.139

## Data Availability

Not applicable.

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
