# Peer review of "A Hybrid Dragonfly Algorithm for Efficiency Optimization of Induction Motors"

_sensors, 2022, doi:10.3390/s22072594_

Round 1

Reviewer 1 Report

  1. The manuscript is concerned with a hybrid dragonfly algorithm for the efficiency optimization of induction motors. It is relevant and within the scope of the journal.
  2. However, the manuscript, in its present form, contains several weaknesses. Adequate revisions to the following points should be undertaken in order to justify recommendations for publication.
  3. For readers to quickly catch the contribution in this work, it would be better to highlight major difficulties and challenges, and your original achievements to overcome them, in a clearer way in the abstract and introduction.
  4. The scrounger strikes Levy-based dragonfly algorithm was used for the optimal tuning of gains of PI controller. What are the other feasible alternatives? What are the advantages of adopting these soft computing techniques over others in this case? How will this affect the results? More details should be furnished.
  5. The proportional gain and integral gain were considered. What about the differential part? What are the advantages of adopting these terms over others in this case? How will this affect the results? More details should be furnished.
  6. The contributions and novelty of the paper should be added before the last paragraph in the introduction section.
  7. The organization of the rest of the paper should be written as the last paragraph in the introduction section.
  8. The future scope should be added to the conclusion section.
  9. The authors did any conduct any numerical benchmarking for their hybrid algorithm. The authors are requested to adopt one benchmark suite of CEC test sets and conduct the necessary benchmarking.
  10. The benchmarking should compare the proposed algorithm with other algorithms such as sine cosine algorithm, multiverse algorithm, grey wolf optimizer, moth flame algorithm, grasshopper algorithm, etc.
  11. The parametric and non-parametric comparison should be made.
  12. Some assumptions are stated in various sections. More justifications should be provided on these assumptions. Evaluation on how they will affect the results should be made.
  13. The performance of the proposed method was examined in terms of power efficiency, speed and torque under four configurations like speed 90 and torque 130, speed 120 and torque 150, speed 150 and torque 180, and speed 180 and torque 200. What are the other feasible alternatives? What are the advantages of adopting these parameters over others in this case? How will this affect the results? More details should be furnished.
  14. The discussion in the present form is relatively weak and should be strengthened with more details and justifications. A separate discussion section should be added to the manuscript.
  15. The proposed algorithm was tested using other data in references [25] and [26] which is not enough. Comparison with other state-of-the-art algorithms like sine cosine algorithm, multiverse algorithm, grey wolf optimizer, moth flame algorithm, etc.

Author Response

Reviewer 1: 

The manuscript is concerned with a hybrid dragonfly algorithm for the efficiency optimization of induction motors. It is relevant and within the scope of the journal. However, the manuscript, in its present form, contains several weaknesses. Adequate revisions to the following points should be undertaken in order to justify recommendations for publication.

  1. For readers to quickly catch the contribution in this work, it would be better to highlight major difficulties and challenges, and your original achievements to overcome them, in a clearer way in the abstract and introduction.

   Ans: The modifications have been done in the abstract and introduction part which highlights   

       the difficulties and challenges. After that a clear objective of the work has been

       highlighted by using proposed method which enhances the efficiency.

  1. The scrounger strikes Levy-based dragonfly algorithm was used for the optimal tuning of gains of PI controller. What are the other feasible alternatives? What are the advantages of adopting these soft computing techniques over others in this case? How will this affect the results? More details should be furnished.

   Ans: The other feasible techniques are GA, PSO and many meta-heuristic algorithms. By  

             adopting these soft computing techniques, more optimize results can be obtained.

             Meta-heuristic algorithms are based on animal search behavior and it solves the

             continuous optimization problems. The proposed SL-DA algorithm shows better

             optimization process as compared to conventional methods of reference [26]. All details

             have mentioned.

  1. The proportional gain and integral gain were considered. What about the differential part? What are the advantages of adopting these terms over others in this case? How will this affect the results? More details should be furnished.

  Ans:  The proportional and Integral gain of all the methods are presented in Table 5. The PI

            controller gain is used for the tuning purpose and its response is high. The derivative

            control normally slows down the response of the circuit, hence not used. In this paper

            only PI controller is used. The extended details are presented in Section 4.1.

  1. The contributions and novelty of the paper should be added before the last paragraph in the introduction section.

 Ans:   Included in the introduction section and a slight information has been also presented in

            section 4.1 which deals in tuning of PI controller.

  1. The organization of the rest of the paper should be written as the last paragraph in the introduction section.

 Ans:   Included in the last paragraph of the introduction section.

  1. The future scope should be added to the conclusion section.

Ans:     Included in the conclusion part of section 6 as per reviewer suggestion.

  1. The authors did any conduct any numerical benchmarking for their hybrid algorithm. The authors are requested to adopt one benchmark suite of CEC test sets and conduct the necessary benchmarking.

Ans:    No numerical benchmarking has done; because this paper only focused on tuning of PI             controller which enhances drive performance. For this, only base paper [26] data has been included and compared with the proposed SL-DA algorithm.

  1. The benchmarking should compare the proposed algorithm with other algorithms such as sine cosine algorithm, multiverse algorithm, grey wolf optimizer, moth flame algorithm, grasshopper algorithm, etc.

Ans:    Thanks for your kind comment, but we hope you will consider that this is an application-oriented paper and he Congress on Evolutionary Computation (CEC) test cases can be a part of future  work for analyzing the performance of induction motor drive.

  1. The parametric and non-parametric comparison should be made.

 Ans:   The scope of the paper is to enhance the performance of drive like enhancing efficiency

            by tuning PI controllers. Therefore, no parametric and non-parametric comparison has

           done and the comparison is only based on the accuracy. The results are compared with the base paper [26], and we hope you will find it satisfactory.

  1. Some assumptions are stated in various sections. More justifications should be provided on these assumptions. Evaluation on how they will affect the results should be made.

Ans:    Only the load condition is varied and other parameters are kept constant so that result are

            justified according to the ideal system behavior.

  1. The performance of the proposed method was examined in terms of power efficiency, speed and torque under four configurations like speed 90 and torque 130, speed 120 and torque 150, speed 150 and torque 180, and speed 180 and torque 200. What are the other feasible alternatives? What are the advantages of adopting these parameters over others in this case? How will this affect the results? More details should be furnished.

Ans:   The performance evaluation is done under different rated conditions of speed and torques

            i.e. at rated condition, below rated condition and for above rated value to analyse the

            overall performance of the drive.

  1. The discussion in the present form is relatively weak and should be strengthened with more details and justifications. A separate discussion section should be added to the manuscript.

Ans:    The discussion of each result has been given separately in sub-sections 5.1, 5.2…..5.7 with

            their tables and data. The discussion has done according to data presented in the tables.

  1. The proposed algorithm was tested using other data in references [25] and [26] which is not enough. Comparison with other state-of-the-art algorithms like sine cosine algorithm, multiverse algorithm, grey wolf optimizer, moth flame algorithm, etc.

Ans:    Thanks for your comment, and we surely consider this in our future studies.

Reviewer 2 Report

The paper deals with novel algorithms for efficiency optimization of induction motors. Clearly, this is an emerging topic with clear societal impact. Moreover, the topic is in the scope of the Journal. The structure of the paper is quite understandable and the language quality appropriate for the journal. However, there are certain points that can be improved:

1) Still there are some grammar errors (e.g. "Induction motors tends ..." in abstract). So I propose the revise grammar and spelling again.
2) The argument that "electricity is wasted by motors, which ends in increased computational cost" sounds a bit strange to me and is not properly justified.
3) Also emphasizing constant speed applications does not make sense. I think this is relevant e.g. for belts, but not for robots and mechatronic systems.
4) Some section titles could be more specific (e.g. "Methodology" is quite general).
5) I propose to doublecheck texts in all Figures, e.g. first block in Figure 1 refers to "current regula-".
6) There are lots of frequency domain methods for optimal PID controller design e.g. based on robustness regions. Perhaps they should be mentioned and referred. Brief comparison with the proposed approach would be nice.
7) The title of Section 3.1 (i.e. "General block representation") does not really match with the content of the section.
8) Section 3.2 contains just one sentence, perhaps the text should be restructured somehow.
9) Looking at Fig. 3, there is a question whether signal saturation are considered in the PI control block. 
10) I would improve the format of Figure 5.
11) A bit more ideas of future research can be added to conclusions.
12) The number of references is adequate, however, there are no links to authors previous works. Hence is hard to identify the research baseline.
13) I would unify notation of symbols (e.g. Theta vs Teta), all symbols should be doublechecked.
14) I think it is not necessary to provide ki and kp parameters in 5-digit precision (Table 5). 
15) Several iterative optimization methods are presented. Is there some guarantee that optimal setting is really found? Iterative methods often find just suboptimal solutions 
16) I think the efficiency should be analysed in context of final application (i.e. where the motor is mounted), then different tuning rules and controller parameters can be optimal. Also lots of other parameters are entering the game to make the whole machine optimal. More comments on that issue would be nice.

There are also some positive aspects, e.g focus on energy optimization (high impact topic) and detailed analysis of results.

Despite above mentioned comments, the paper can be re-considered after revision.

Author Response

Reviewer 2:

The paper deals with novel algorithms for efficiency optimization of induction motors. Clearly, this is an emerging topic with clear societal impact. Moreover, the topic is in the scope of the Journal. The structure of the paper is quite understandable and the language quality appropriate for the journal. However, there are certain points that can be improved:

  • Still there are some grammar errors (e.g. "Induction motors tends ..." in abstract). So I

      propose the revise grammar and spelling again.

Ans: The grammar has been corrected in the abstract as suggested.

  • The argument that "electricity is wasted by motors, which ends in increased

      computational cost" sounds a bit strange to me and is not properly justified.

Ans: The complete portion of this sentence has been modified as per reviewer suggestions.

3) Also emphasizing constant speed applications does not make sense. I think this is relevant e.g. for belts, but not for robots and mechatronic systems.

Ans: The complete work has been analyzed for induction motor which is driving a load for a  

          particular application whose speed and torque are fixed.

4) Some section titles could be more specific (e.g. "Methodology" is quite general).

Ans: It has been replaced with “Proposed Methodology”

5) I propose to doublecheck texts in all Figures, e.g. first block in Figure 1 refers to "current regula-".

Ans.  Revised as suggested.

6) There are lots of frequency domain methods for optimal PID controller design e.g. based on robustness regions. Perhaps they should be mentioned and referred. Brief comparison with the proposed approach would be nice.

Ans:   More papers and contents are added in the papers.

7) The title of Section 3.1 (i.e. "General block representation") does not really match with the content of the section.

Ans: The contents of sub-ection 3.1 is modified according to the title.

8) Section 3.2 contains just one sentence, perhaps the text should be restructured somehow.

Ans: Modified and merged in sub-ection 3.1.

9) Looking at Fig. 3, there is a question whether signal saturation are considered in the PI control block. 

Ans:  No, signal saturation are not considered.

10) I would improve the format of Figure 5.

      Ans: Corrected and done

11) A bit more ideas of future research can be added to conclusions.

      Ans:  Added in the conclusion part.

12) The number of references is adequate, however, there are no links to authors previous works. Hence is hard to identify the research baseline.

      Ans: Included more papers in reference section related to authors previous work.

13) I would unify notation of symbols (e.g. Theta vs Teta), all symbols should be double checked.

      Ans:  Revised as suggested.

14) I think it is not necessary to provide ki and kp parameters in 5-digit precision (Table 5). 

      Ans:   The values of kp and ki parameters have corrected up to 3 digits.

15) Several iterative optimization methods are presented. Is there some guarantee that optimal setting is really found? Iterative methods often find just suboptimal solutions 

Ans: As referred to table5, the three techniques are compared for PI controller and efficiency

         are computed. When a comparison among all techniques are made, the proposed method

         show superiority over others. In terms of efficiency, output power and kp and ki values.

         Hence, our motive is achieved which guarantees the optimal condition of proposed

         controller.

16)   I think the efficiency should be analysed in context of final application (i.e. where the motor is mounted), then different tuning rules and controller parameters can be optimal. Also lots of other parameters are entering the game to make the whole machine optimal. More comments on that issue would be nice.

Ans: The paper mainly focused on the effect of PI controller to evaluate the efficiency of the

         drive. Other internal parameters of the machine are kept constant for each case study. The

         different tuning rules are used; from which the proposed model of SL-DA algorithm

         shows the superiority over other controllers.

Round 2

Reviewer 1 Report

The paper can be accepted in the present form.

Author Response

All suggestions of the reviewer are implemented.

Reviewer 2 Report

The second version of the paper has significantly higher quality. The authors responded properly to all reviewer comments. The paper can be accepted provided that all reviewers agree and after final formatting.

Author Response

As per suggestion, I have modified the manuscript.